# The Gaps in Health-Adjusted Life Years (HALE) by Income and Region in Korea: A National Representative Bigdata Analysis

**DOI:** 10.3390/ijerph18073473

**Published:** 2021-03-27

**Authors:** Young-Eun Kim, Yoon-Sun Jung, Minsu Ock, Hyesook Park, Ki-Beom Kim, Dun-Sol Go, Seok-Jun Yoon

**Affiliations:** 1Department of Big Data Strategy, National Health Insurance Service, 32 Geongang-ro, Wonju 26464, Korea; kimyes4454@gmail.com; 2Department of Public Health, Graduate School, Korea University, 73, Goryeodae-ro, Seongbuk-gu, Seoul 02841, Korea; sunnyaurora@nate.com (Y.-S.J.); socommat@korea.ac.kr (K.-B.K.); 3Department of Preventive Medicine, University of Ulsan College of Medicine, 88 Olympic-ro 43-gil, Songpa-Gu, Seoul 05505, Korea; ohohoms@naver.com; 4Department of Preventive Medicine, Ulsan University Hospital, University of Ulsan College of Medicine, 877 Bangeojinsunhwando-ro, Dong-gu, Ulsan 44033, Korea; 5Department of Preventive Medicine, Graduate Program in System Health Science and Engineering, Ewha Womans University, College of Medicine, 25 Magokdong-ro 2-gil, Gangseo-gu, Seoul 07804, Korea; hpark@ewha.ac.kr; 6Department of Health Care Policy Research, Korea Institute for Health and Social Affairs, Sejong 30147, Korea; kds237@gmail.com; 7Department of Preventive Medicine, Korea University College of Medicine, 73, Goryeodae-ro, Seongbuk-gu, Seoul 02841, Korea

**Keywords:** health-adjusted life years, inequality, years lived with disability, income, region

## Abstract

This study aims to calculate the health-adjusted life years (HALE) by using years lived with disability (YLD) from the national claims data, as well as to identify the differences and inequalities in income level and region. The study was carried out on total population receiving health insurance and medical benefits. We calculated incidence-based YLD for 260 disease groups, and used it as the number of healthy years lost to calculate HALE. We adopted the insurance premium to calculate the income as a proxy indicator. For the region classification, we chose 250 Korean municipal-level administrative districts. Our results revealed that HALE increased from 2008 to 2018. HALE in males increased faster than that in females. HALE was higher in higher income levels. In 2018, the gap in HALE between Q1 and Q2, the lower income group, was about 5.57 years. The gap in females by income level was smaller than that in males. Moreover, the gap in HALE by region was found to increase. Results suggest that there is an inequality in YLD in terms of income level in Korea. Therefore, we need intensive management for the low-income group to promote HALE at the national level.

## 1. Background

Health span is the length of time in one’s life during which an individual is in a healthy state without any physical or mental disability [1]. In general, it is calculated by subtracting the period of disease prevalence from life expectancy. Various calculation methodologies exist according to the definition and calculation method of “health.” In particular, disability-free life expectancy (DFLE), healthy life expectancy (HLE), health-adjusted life expectancy (HALE), and quality-adjusted life expectancy (QALE) are representative indicators of the healthy lifespan of a specific population. Among various indicators of population health, HALE summarizes the expected number of years to be lived in what might be termed the equivalent of “full health” [2]. Some consider that HALE provides the best available summary measure for measuring the overall level of health for a population [3]. In this context, the World Health Organization (WHO) has used it as an official indicator of annual reports to provide information about the average level of the population of member states [2,3]. The WHO calculated HALE based on years lived with disability (YLD), a component of disability-adjusted life years (DALY), produced in the Global Burden of Disease (GBD) study by Institute for Health Metrics and Evaluation (IHME) [4].

The HALE is an important indicator to establish the National Health Plan and other national-level health policies. The GBD Study in IHME calculates disease burdens of countries by gender, age, and illness through “estimation” based on published papers and other estimation methods for meta-regression [5]. This means that epidemiologic parameters such as prevalence, incidence, and fatality may not have been measured coherently. The accuracy of epidemiological indicators would have been high if there were many related studies or qualitatively excellent studies, but on the contrary, it often relies on subjective questions or surveys. In this context, the accuracy and consistency of the data sources was critical in calculating HALE.

Previous studies have calculated HALE for Korean individuals [6,7]. However, since they used the National Health and Nutrition Examination Survey data to measure the health loss, it has the limitation that only a narrow spectrum of diseases (45 diseases) was included in it. Additionally, it did not measure the socioeconomic health inequality. Broadening the spectrum of diseases is necessary to calculate accurate years of healthy life lost. We also need the review on data sources to obtain the reliability and coherence. 

This study aims to calculate HALE by using YLD from national claims data as well as to identify the differences in income level and region.

## 2. Materials and Methods

### 2.1. Data and Units of Analysis

South Korea provides Universal Health Care (UHC) through national health insurance and public health benefit to all its nationals. As of January 2020, over 52 million people had received healthcare benefit, 97.2% of beneficiaries were under health insurance coverage, and 2.8% of people were public health beneficiaries. As a compulsory social insurance, the Korean health insurance covers the whole population living in the country and the NHIS (National Health Insurance Service) covered 63.8% of all medical expenses in 2018 [8,9]. The NHIS claims database contains information about beneficiaries’ personal information such as insurance premium, demographics, medical uses, diagnosis, care start and end date, prescription details, and operation (surgery) details excluding out-of-pocket payment [10]. Therefore, to measure the YLD in Korea, we calculated the prevalence and incidence rates for 260 disease groups, which are main input values for YLD, using the claims data. In case of injuries, we used National Hospital Discharge Survey by Korea Centers for Disease Control and Prevention (KCDC) to supplement the accuracy of the claims data [11]. Additionally, the cause of death and life table were obtained from the life table published by Statistics Korea [12]. Duration of disease and the average age at onset were estimated using the DISMOD-2 program by WHO, and infectious diseases such as influenza and varicella were directly calculated using the claims data [13]. Disability weight calculated for Koreans was also used [14]. We measured YLD using an incidence-based approach by considering the prevalence, incidence, fatal, mortality, duration of disease, and disability weight for 260 disease groups, and this was used as the number of healthy years lost to calculate HALE. The list of 260 causes, and other detailed methodologies, were referenced from the methodology developed in the Korean National Burden of Disease study [15].

We adopted the insurance premium to calculate the income as a proxy indicator. The NHIS scales insurance premiums based on subscribers’ wages and incomes. Therefore, we used an equivalized annual household income based on insurance premium in this study and divided it into quintiles by year and gender. Equivalized annual household income is derived as follows:$$Equivalized\;Annual\;Household\;Income\;=\;\frac{Annual\;Household\;Income}{Number\;of\;Household\;Member^{0.5}}$$

For the region classification, we chose 250 Korean municipal-level administrative districts, consisting of 67 cities (“Si”), 114 counties (“Gun”), and 69 districts (“Gu”), to cover the whole country. In South Korea, the planning and implementation of health promotion programs to increase healthy life expectancy are carried out in Si, Gun, and Gu, and we therefore determined that considering municipal-level administrative divisions was optimal for understanding the differences in regional health levels.

### 2.2. Statistical Analysis

The sum of YLD is mutually independently calculated for disease groups. However, if comorbidity exists, the sum of YLD causes a problem of overestimation. To reflect the loss of YLD due to comorbidity disease in the equation, we utilized the Monte-Carlo simulation to estimate the loss and finally calculated the HALE. This method was used in GBD 2016 [16]. We generated 40,000 simulants and repeated the procedure 1000 times for each sex and age group to calculate YLD reduction percentage.

To measure HALE in Korea, we used the Sullivan method [17]. It is expressed as the difference between life expectancy calculated from the whole population, mortality by age group, and the years of healthy life lost derived from YLD. The WHO and IHME use the same method to estimate HALE [4,18]. We calculated HALE from 2008 to 2018 depending on gender, income level, and region. In this study, to minimize the effect of outliers, we considered only the difference between the 95th and 5th percentile, while comparing the difference in HALE by regions. Finally, we tested the statistical significance of the HALE trend through the generalized linear model
(GLM), and predicted the HALE of the total population, males, and females in 2030.

We used SAS version 9.4 (SAS Institute Inc., Cary, NC, USA) for statistical analysis in this study.

## 3. Results

### 3.1. HALE in Korea, by Gender, Income Level, and Year

The HALE in Korea was 68.89 years in 2008, increasing to 70.43 years in 2018, an increase of 1.54 years over 10 years (average 0.15 year per year). When we look at HALE results by gender in 2008, HALE for males was 66.47 years, and it increased by approximately 1.78 years to 68.25 years in 2018, while for females, it increased by 1.37 years from 71.00 in 2008 to 72.37 in 2018. Our results reveal that males had a faster HALE increase rate than females during 2008–2018. The regression analysis results showed that the HALE for the overall population (t-value = 7.27), males (t-value = 7.94), and females (t-value = 6.91) increased statistically significantly by year (*p* < 0.05). We identified a decreasing trend in the gap between females and males: from 4.53 years in 2008, to 4.12 years in 2018, which was statistically significant (t-value = −7.67, *p* < 0.05). In the case of HALE by income level, the gap between the 1st quintile and 5th quintile based on income level decreased between 2008 and 2012 from 7.94 to 6.72 years; after 2012, it increased to 8.04 years in 2018. HALE in Q1 increased by 1.42 years from 63.82 in 2008, to 65.24 in 2018 (t-value = 5.63, *p* < 0.05), while in Q2, it increased by 1.30 years (t-value = 6.05, *p* < 0.05), Q3 by 1.87 years (t-value = 8.02, *p* < 0.05), and Q4 by 1.70 years (t-value = 7.13, *p* < 0.05). In Q5, within the highest income level group, it increased by approximately 1.53 years from 71.76 years in 2008 to 73.28 years in 2018 (t-value = 5.38, *p* < 0.05). Our results reveal that the HALE for each income level showed a statistically significant increasing trend by year, and that HALE increases with an increase in income level. The gap between Q1, the lowest income group, and Q2, the second lowest, was approximately 5.57 years in 2018 (Table 1). According to the results of our analysis using the GLM model, the HALE in 2030 is predicted to be 73.3 for the total population, 71.4 for males, and 75.0 for females.

### 3.2. HALE by Income Level for Males and Females in Korea

HALE by gender and income level showed that HALE increased from 2008 to 2018, for all income levels, and for both males and females. Identified trends by year were as follows: For males, at Q1, the lowest income level, HALE was 61.09 years in 2008, and gradually increased by 1.35 years to 62.44 years in 2018. At Q5, the highest income level, HALE increased by 1.86 years from 69.73 years in 2008 to 71.59 years in 2018. For females, at Q1, HALE was 66.60 years in 2008, and gradually increased by 1.63 years to 68.23 years in 2018. At Q5, HALE increased by 1.24 years from 73.29 years in 2008 to 74.53 years in 2018. The increase in HALE for all income levels for both males and females was statistically significant (*p* < 0.05). For males, the gap in HALE according to income level increased by 0.51 years from 8.64 years in 2008 to 9.15 years in 2018. On the other hand, for females this gap decreased from 6.70 years in 2008 to 6.30 years in 2018, while males have a greater gap in HALE by income level than females. That is, inequality in HALE according to income level is considered to be greater in males. However, in both males and females, the increase in the HALE difference according to income levels was not statistically significant by year (Figure 1). 

### 3.3. Distribution of the HALE by Region

Results of HALE according to 250 administrative regions are as follows. The gap between 95th and 5th percentile increased by 0.82 years, from 4.88 years in 2008 to 5.70 years in 2018. According to the gender classification, the gap in HALE for males increased by 0.77 years from 5.93 years in 2008 to 6.71 years in 2018, while for females, it increased by 0.54 years from 4.80 years in 2008 to 5.34 years in 2018. The gap in HALE by region slightly increased and decreased every year, and in particular, it showed a decreasing trend from 2016, but showed a tendency to increase over the entire 10 years. According to the results of the regression analysis, the gap in HALE by region in the total population, males, and females increased statistically significantly (Figure 2).

## 4. Discussion

This study presents HALE based on YLD calculated from 2008 to 2018 using national claims data. We computed the gap in HALE by gender and income level to measure the equity of HALE. HALE increased from 2008 to 2018, and results by gender suggest that HALE in males increases faster than that in females. In terms of gender differences, females generally have higher morbidity and are more interested in medical care than males. In addition, as women’s work–family conflict increases significantly over time, responsibilities regarding employment, housework, pregnancy, childbirth, and childrearing would also increase, which would directly or indirectly affect females’ health. Results by income level reveal that HALE is higher in higher income levels. The gap in HALE between Q1 and Q2, the lower income group, was about 5.57 years in 2018. The gap by income level in females was smaller than in males, which can be because males have greater inequality in terms of HALE according to income level. This is similar to the results derived from another study [19] that compared the gap in healthy life expectancy according to income level by gender, using the subjective health assessment life expectancy. This suggests that males’ health level may be more sensitive to the socioeconomic level than females’ health level. The gap in HALE between regions has increased overall over time.

The gap in life expectancy by gender decreased from 6.55 years in 2008 to 6.03 years in 2017 [20]. The gap in life expectancy by income level also decreased from 6.90 years in 2008 to 6.48 years in 2017. According to the results, the gap in gender has been on the decline, from 4.53 in 2008 to 4.12 in 2018, and the gap based on income levels decreased from 7.94 in 2008 to 6.72 in 2012, thereafter increasing to 8.04 years in 2018. The lowest income level group, especially, has lower HALE compared to other quintiles. Several studies show that smoking, alcohol usage, and other health behaviors of the low-income bracket are worse than in other brackets [21,22,23]. We need intensive management for the low-income group to promote HALE at the national level. Research is needed on how to set goals to manage health hazards by calculating population attributable fractions (PAF), a fraction that major health hazards such as smoking, drinking, and obesity contributed to YLD, and improving HALE by preventing certain diseases. 

This study suggests that the regional gap in HALE is increasing overall. HALE has decreased since 2016 among the entire period of the study, from 2008 to 2018, but it is necessary to continuously observe the trend of increase and decrease after more time series data are accumulated. South Korea has the lowest birthrate and the fastest aging population rate in the world, and the income gap between regions is also deepening. The gap in HALE by regions is correlated with socioeconomic characteristics, smoking, and health-infrastructure-related indicators [7]. Therefore, there is a need for further research to identify the origin of the regional HALE gap. In particular, research is needed that reveals the association of the distribution of infrastructure with HALE. As research reveals this association, we can expect to improve health levels in regions with low HALE by preparing the basis for a balanced medical resource allocation policy. 

We calculated years lived in poor health from the difference between life expectancy in Korea, suggested in Khang et al., and HALE from this study [20]. In terms of the “years lived in poor health”, we found an increase of 1.32 years from 11.15 years in 2008 to 12.47 years in 2017. For males, there was an increase of 1.39 years from 10.04 years in 2008 to 11.43 years in 2017. For females, there was an increase of 1.26 years from 12.06 years in 2008 to 13.32 years in 2017. The difference in life expectancy by gender was 6.03 years in 2017, but the difference in HALE was smaller, at 4.14 years in the same period. That is, females have relatively more “years lived in poor health” than males. In 2017, Q1 spent 13.29 years, while Q5 spent 12.22 years in terms of time spent in an unhealthy state, and the difference between the two groups was 1.07 years. The results suggest that there is an inequality in “years lived in poor health” in terms of income level in Korea. 

South Korea’s HALE in 2016, from GBD 2016, was 68.49 years for males and 72.97 years for females, and “years lived in poor health” was 11.25 years for males and 9.18 years for females [24]. The fact that females have a higher HALE and higher “years lived in poor health” than males matched the results of this study. Moreover, the derived HALE in this study (70.56 years in 2016) is similar to the HALE derived in GBD 2016 (70.76 years in 2016). However, South Korea’s HALE in 2016, from GBD 2017, is 71.57 years (69.59 years for males, 73.37 years for females), and it increased by about 0.81 years from GBD 2016 [24]. This difference between the GBD Study and this study is caused by the adoption of the procedure of input variable estimation by WHO and IHME.

The WHO also calculates HALE based on YLD from the GBD Study. The WHO currently presents South Korea’s HALE in 2019 as 73.1 years (71.3 years for males, 74.7 years for females). In 2015, the WHO’s HALE for South Korea was 72.0 years (70.2 years for males, 73.7 years for females) [25]. It was updated based on World Health Statistics 2020 on December 4, 2020, but South Korea’s HALE in 2015 was presented as 73.2 years using World Health Statistics 2016 [26]. The limitations mentioned above make it difficult for individual countries to develop and evaluate the health goal using HALE as a national indicator. Therefore, when establishing and evaluating Korea’s policy, data-driven YLD is more compatible than model-driven YLD, which the WHO and IHME have adopted. Additionally, HALE from the WHO and IHME can be used as a powerful indicator for a comparative study between countries, but it is not being produced in terms of equity, such as income in individual countries or regional differences in HALE.

In Korea, the national Health Plan (HP) is established every 10 years, and aims to increase healthy life expectancy and achieve health equity. Most recently, in January 2021, the National Health Plan 2030 (HP2030) was announced. In the past, Health Plan 2020 (HP2020) used the WHO’s HALE to set the target for Koreans’ health until 2020, but there were many difficulties in evaluating the achievement of policy outcomes due to uncertainty in the WHO’s calculation cycle for HALE. Accordingly, in HP 2030, it was decided to set a target value for future Koreans’ healthy life expectancy based on HALE calculated in Korea. In addition, as revealed in this study’s results, it was possible to calculate the difference in HALE between income levels and regions through domestic data. Based on this result, the target value of “achieving health equity,” another of the overall goals, could be set. The results of this study are meaningful since they can be used as a scientific and useful indicator in the future evaluation and monitoring of health policy performance.

This study has some limitations. When calculating HALE in a small area, some outliers occurred in HALE due to the lack of population and deaths. In future, it will be necessary to develop a methodology for HALE calculation in small areas. In addition, in the process of calculating HALE, we used Monte-Carlo simulation to reflect the YLD loss due to comorbidities. In future studies, it is necessary to further improve the methodological completion by estimating the amount of reduction in YLD due to comorbidities using data for the entire population. In this study, equivalized annual household income was used, but it is necessary to discuss whether household income is identified with individual income. However, through the study, we were able to verify HALE’s capability as a measure of equity. This is the first HALE study derived from national data for the whole population.

## 5. Conclusions

In summary, our study showed the HALE in Koreans using national representative big data 2008–2018. We identified the existence of inequality in HALE according to the region and income, and that inequality between regions and income levels in HALE keeps increasing as time goes on. Therefore, we need intensive management for the low-income group to promote HALE at the national level.

## Figures and Tables

**Figure 1 ijerph-18-03473-f001:**
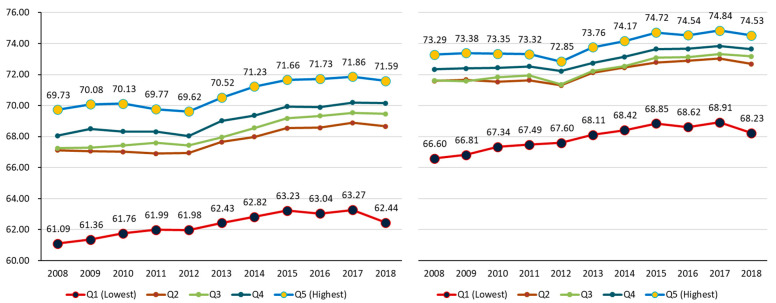
The HALE by income level for males and females in Korea (2008–2018) (unit: years).

**Figure 2 ijerph-18-03473-f002:**
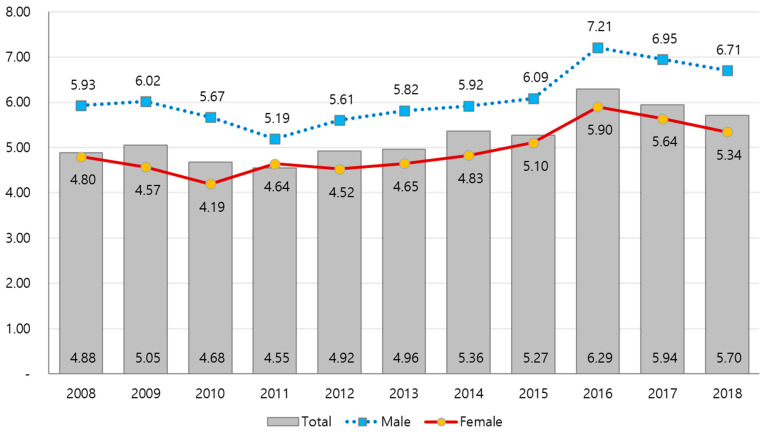
Difference (95th and 5th percentile) of HALE for males and females by region in Korea (unit: years).

**Table 1 ijerph-18-03473-t001:** Annual health-adjusted life years (HALE) (unit: years) by gender and income level in Korea (2008–2018).

	2008	2009	2010	2011	2012	2013	2014	2015	2016	2017	2018
**Overall population**	68.89	69.05	69.16	69.20	68.97	69.69	70.10	70.59	70.56	70.78	70.43
**Gender**											
Male	66.47	66.69	66.78	66.77	66.66	67.36	67.82	68.35	68.33	68.58	68.25
Female	71.00	71.09	71.24	71.32	71.00	71.72	72.08	72.54	72.49	72.72	72.37
Difference (Female-Male)	4.53	4.40	4.46	4.54	4.34	4.36	4.26	4.20	4.16	4.14	4.12
**Income level**											
Q1 (Lowest)	63.82	64.06	64.53	64.72	64.76	65.25	65.58	66.00	65.78	66.03	65.24
Q2	69.52	69.53	69.43	69.42	69.25	70.03	70.37	70.80	70.89	71.10	70.81
Q3	69.61	69.61	69.81	69.94	69.56	70.25	70.73	71.30	71.40	71.59	71.48
Q4	70.39	70.66	70.59	70.63	70.33	71.08	71.47	72.00	71.98	72.21	72.10
Q5 (Highest)	71.76	72.01	72.00	71.80	71.48	72.39	72.93	73.44	73.37	73.58	73.28
Difference (Q5-Q1)	7.94	7.95	7.47	7.08	6.72	7.14	7.35	7.44	7.58	7.55	8.04

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
