# Peer review of "The Gaps in Health-Adjusted Life Years (HALE) by Income and Region in Korea: A National Representative Bigdata Analysis"

_ijerph, 2021, doi:10.3390/ijerph18073473_

Round 1
Reviewer 1 Report
Health-adjusted life expectancy (HALE) years.
Healthy Life Expectancy (HLE) and Years of Disability (YLD). In addition, AVACs play a key role today as outcome measures in economic evaluation studies (cost-effectiveness and cost-benefit analysis) and (cost-effectiveness and cost-utility analysis) in measuring the impact of actions. or specific interventions to reverse the load. of the disease and improve the quality of life and life expectancy of patients. Its generic nature makes it possible to compare the impact of health interventions between different diseases and in this case between regions and other determinants of health.
However, I believe it is important to introduce several points by way of argumentation related to the definition of health which is commonly defined as a "state of complete well-being" which would not be adequate given the increase in chronic diseases and mental health illnesses.
On the other hand, women are increasingly becoming part of the labor force and now equal or exceed the incidence of occupational and professional accidents. In addition, it is known that in some societies, to reach certain standards of living, it has implied an increase in the recommended working hours (work presentiment and the double role that women play to a greater extent, work stress, sleep, and other problems).
Inequalities in care could influence overestimation of health spending in higher-income populations. A person with greater resources tends to have higher health spending than a person with a lower socioeconomic level.
This could overestimate or underestimate some results. Because of the above, I suggest devoting some thought to this aspect. However, this is only a suggestion. In the estimation process some experts establish the existence of controversial aspects regarding the selection of the life table, the difference between the life expectancies of men and women, the temporal proximity to the measurement and the weighting of the severity of the disability states. To better understand these controversies, if tables with high life expectancy are selected, a weighting bias of years due to premature death is generated in individuals who die according to another reality. Therefore, it is important to raise this argument in the discussion of the results.
From the methodological point of view, it is not clear how the overestimation is controlled with the Monte Carlos simulation and what would be the statistics to consider an adequate model (in this case, I consider it important to establish this in the section of statistical analysis). Among the disadvantages we can mention that many variables improve the estimates, however, the simulation does not generate optimal situations or solutions, each simulation is unique. Therefore, Monte Carlos does not allow to look for the best solution but to evaluate different alternatives. Therefore, it is important to explain a little more how this technique helped for the estimates.
From line 165-178 the same results are repeated. In this case it is convenient to make a discussion of the result and to contract the differences found with other studies, with different or similar realities. This result should be discussed.

Author Response
"Please see the attachment."

Reviewer 2 Report
- The calculation of HALE by Sullivan's method needs to be shown in the paper. not understood to the readers what was this method and how did they calculated the HALE.
- HALE was disaggregated by 250 Administrative units. It is not clear in the description section that how the HALE was varying among these regions. Make some regional blocks and then show the statistical difference between the regional blocks. Otherwise, the regional results are not explicit.
- The results were not prepared based on the statistical inference, especially by gender, income level, and regions. Statistical significance of the mean differences needs to be shown.
- The use of the regression method is needed to get the expected results. Only seeing the trend, it is not the scientific results.
Author Response
"Please see the attachment."
